CERN-TH-2024-223

# Fractional Hydrodynamic Transport from the Witten Anomaly

Joe Davighi

*Theoretical Physics Department, CERN, 1211 Geneva 23, Switzerland*[*]

Nakarin Lohitsiri

*Department of Mathematical Sciences,*
*Durham University, Upper Mountjoy, Stockton Road,*
*Durham, DH1 3LE, United Kingdom*[†]

Napat Poovuttikul

*High Energy Physics Research Unit, Department of Physics,*
*Faculty of Science, Chulalongkorn University, Bangkok 10330, Thailand*[‡]

We study the physical consequences of 't Hooft anomalies in the high-temperature limit of relativistic quantum field theories with $SU(2)$, or more generally $USp(2N)$, global symmetry. The global anomaly afflicting these symmetry groups results in new transport phenomena akin to the chiral magnetic and chiral vortical effect, predicting conductivities that are fractionally quantised in units of a half, reflecting the 2-torsion in the bordism groups $\Omega_5^{\mathrm{Spin}}(BUSp(2N)) \cong \mathbb{Z}_2$.

[*] joseph.davighi@cern.ch

[†] nl313@cantab.ac.uk

[‡] napat.po@chula.ac.th

## CONTENTS

## I. INTRODUCTION

In the game of building effective field theories (EFT), understanding the symmetry structure is of paramount importance. This is particularly true when the symmetry is anomalous [1]. An anomalous symmetry encodes robust information that remains unchanged along the renormalisation group flow from the ultraviolet (UV) to the infrared (IR). This means that an IR EFT must have the same anomaly as the microscopic UV theory it descends from.

Important applications of this 'anomaly matching' idea date back to the early years following the discovery of the chiral anomaly by Adler [2], Bell and Jackiw [3]. The Wess–Zumino–Witten (WZW) term [4, 5] in the chiral Lagrangian EFT describing pions, that was proposed to match anomalies in the underlying theory of quarks and gluons, has many observable consequences: it is needed to explain the rapid decay of neutral pions to photons, and that the $\phi$ meson can decay to both $K^+K^-$ and $\pi^0\pi^+\pi^-$ final states. The phenomenological importance of WZW terms that match anomalies persists for quantum systems at finite temperature, where the non-conservation of classically-conserved currents leads to the

phenomena of *anomaly induced transport*. Examples that have been tested in the laboratory include the chiral magnetic effect and related phenomena [6–9]. We refer the reader to Ref. [10] for a recent review, and [11, 12] concerning the experimental observation of these effects.

All these finite-temperature examples concern the physics of *perturbative anomalies* (also known as *local anomalies*), namely the violation of classical conservation laws associated with continuous would-be Noether currents that can be computed from 1-loop Feynman diagrams. Witten discovered that more subtle non-perturbative effects can also render a symmetry anomalous [13], that cannot be seen at all in the perturbative Feynman diagram expansion. Such a *non-perturbative anomaly* (or *global anomaly*) afflicts an SU(2) symmetry in $3 + 1$ dimensions with a massless Weyl fermion in the fundamental representation: the partition function in this case flips sign upon certain SU(2) transformations [13, 14].

A global anomaly does not violate the conservation of continuous Noether currents; rather, the partition function suffers from a discrete phase ambiguity. In the SU(2) case, for instance, that ambiguity is a sign *i.e.* is $\mathbb{Z}_2$-valued. One might therefore expect that global anomalies have no observable quantities in many classes of IR EFTs, such as chiral Lagrangians or those describing thermodynamics and hydrodynamics, whose dynamics is described only by continuous variables. This naïve assumption turns out to be incorrect. Important examples of systems with this type of global anomaly have been found in $1+1$ and $2 + 1$ dimensions, corresponding to the phenomena of *topological insulators* and *topological superconductors* (see [15, 16] for reviews), for which microscopic computations reveal that the global anomaly puts non-trivial constraints on the IR physics.

Anomalies, both local and global, are now understood systematically through the idea of anomaly inflow [17], which identifies anomalies with QFTs in one higher spacetime dimension and with certain non-generic properties such as the anomaly theory being *invertible*. Anomaly theories are thence classified via a cobordism theory [18].[1] The cobordism-based classification of global anomalies applies equivalently (but in a shifted spacetime dimension) to classifying so-called 'symmetry protected topological (SPT) phases' of matter [19, 20], of which the topological insulators/superconductors mentioned above are examples.

The construction of effective actions that match anomalies in the IR has also been addressed more-or-less systematically, using the same tools of invertible field theory and cobordism, at least for a class of chiral Lagrangian-like theories, in [21–23]. The same cannot be said, however, for a broader class of EFTs that describe the thermodynamic and hydrodynamic regimes of such systems. While intensive studies have been carried out for systems with perturbative anomalies in the context of the chiral magnetic effect, only a few examples with global anomalies are known [24–28]. The development of general techniques for analysing global anomalies in this context is warranted, especially given the universality of

---

[1] In this paper, a *bordism group* is an Abelian group whose elements are equivalence classes of manifolds equipped with certain structures (such as a spin structure), where two manifolds are equivalent if they can be connected by an interpolating manifold in one dimension higher. *Co*bordism groups are dual to bordism groups; in particular, we refer to the (shifted) Anderson dual, following Freed–Hopkins [18].

the thermodynamic and hydrodynamic EFT limits, which are also believed to be applicable even in (or rather especially in) the strongly coupled regime. Moreover, understanding how anomalies affect such macroscopic descriptions could provide new pathways towards detecting the presence of anomalies experimentally.

In this work we present a procedure for ascertaining when a global anomaly can be diagnosed in the high temperature, hydrodynamic limit of a relativistic theory in 3+1 dimensions, and what consequences this has for transport phenomena. Such transport aspects are often overlooked both in the condensed matter literature, that has considered the consequences of global anomalies but primarily in lower dimensional systems, and in the quark-gluon plasma community, where intensive studies have been done in $3 + 1$ dimensions but only for perturtubative anomalies. To illustrate our methods we focus on the Witten anomaly for $G = USp(2N)$ symmetry [13], including the case $G = SU(2)$, in the high-temperature, hydrodynamic regime. Of the classical simple Lie groups, it is only these that can exhibit a global anomaly in $3 + 1$ dimensions. Well-known examples of mapping tori that probe the Witten anomaly [13, 14] are inconsistent with passing to the high temperature equilibrium phase, and so a key challenge that we overcome is to find a background geometry that can probe the global anomaly in this phase. We then demonstrate how the non-trivial anomaly is manifest through fractionally quantised conductivity in response to an external non-abelian magnetic field and/or vorticity - see Eqs (6.1). This can be understood through a fractional Chern–Simons term being mandated in the EFT obtained by dimensionally-reducing on the thermal cycle.

The remainder of the manuscript is organised as follows. In Section II, we summarise the modern picture for how global anomalies are classified and detected, and adapt this to the hydrodynamic setting in Section III. We then use this method to determine the anomaly that results in the phase ambiguity of the thermal partition function, starting from a fundamental relativistic theory with the Witten anomaly, in Section IV. In section V, we determine the hydrodynamic effective action that captures this anomaly as well as its observable effect in transport phenomena. Open problems and future directions are discussed in Section VI.

## II. METHODS AND BACKGROUND MATERIAL

Consider a relativistic quantum field theory on a $d$-dimensional spin manifold $X$ with symmetry group $G$ that has global anomalies, and assume that the system at high-temperature reaches thermal equilibrium. Following the approach in [27, 28], one can ask whether the anomaly leaves physical effects in the gradient expansions of the thermal partition function $Z_{\mathcal{T}}[X; A]$, and what these effects are. There are three key steps in answering this question:

(i) **Classifying the global anomaly via bordism.** The global anomaly on $X$, with the background gauge field $A$, is described via inflow by an invertible topological theory $Z_{\mathcal{I}}[Y]$ in $d + 1$ dimensions, and such theories are classified by the spin bordism group

$\Omega_{d+1}^{\mathrm{Spin}}(BG)$ [18]. Computing this bordism group will tell us if the theory can have a global anomaly, and the finest possible phase ambiguity probed by $Z_{\mathcal{T}}[X;A]$.

(ii) **Detecting the global anomaly via a mapping torus.** One needs to find a 'mapping torus' $(Y_{\mathrm{tot}}, A)$ in $d+1$ dimensions that interpolates between field configurations and geometries consistent with the high temperature limit and thermal equilibrium assumption, and on which the invertible theory $-i \log Z_{\mathcal{I}}[Y_{\mathrm{tot}}]$ evaluates to the smallest (mod $2\pi$) amount allowed by the bordism group – which means $(Y_{\mathrm{tot}}, A)$ can be taken as a generator for the bordism group. This step is needed to establish that the global anomaly can indeed be probed in thermal equilibrium, and it is not *a priori* the case given the existence of the global anomaly.

(iii) **Matching the global anomaly via the effective action.** Lastly, to track through the consequences for transport phenomena, we need to express $\log Z_{\mathcal{T}}[X;A]$ as a local effective action $S_{\mathrm{eff}}$ composed out of the background metric and gauge fields, that is again compatible with thermal equilibrium [29] in the high-temperature limit. This involves compactification on the thermal cycle. The action is usually organised order-by-order in the derivative expansions. The $S_{\mathrm{eff}}$ should be invariant under small gauge transformations but not under the large gauge transformation and/or diffeomorphism, as dictated by the global anomaly probed by the mapping torus in step (ii).

In the remainder of this section, we briefly review the bordism classification of global anomalies that justifies step (i). The role of the invertible theory $Z_{\mathcal{I}}$ in determining the anomaly is discussed in Section II A, specialised to chiral fermion anomalies in Section II B. A method to evaluate the $\eta$-invariant for a global anomaly via anomaly interplay is discussed in Appendix A, as it is a somewhat technical diversion. We then turn to steps (ii) and (iii), specialising to the thermal EFT limit which is our focus, in Section III.

## A.   Inflow, locality, and anomaly cancellation

First we review how anomalies in the UV theory, in $d$ spacetime dimensions, are captured by an invertible field theory in $d+1$ dimensions.[2] To start, fix the $d$-dimensional spacetime manifold $X$. If a theory $\mathcal{T}$ has a 't Hooft anomaly in $G$, this means that the partition function $Z_{\mathcal{T}}[X]$ is no longer a $\mathbb{C}$-valued function on the space of background $G$ gauge fields, but is instead a section of a complex line bundle over that space. The curvature of that bundle detects perturbative anomalies while, if this bundle is flat, the holonomy encodes any residual global anomalies. For instance, if the anomaly is due to chiral fermions, then

---

[2] Recall that an invertible field theory in $k$ dimensions [30] is a functor that evaluates to a pure-phase when evaluated on a closed $k$-manifold $Y$, and gives an element in a one-dimensional Hilbert space $\mathcal{H}(X)$ when evaluated on a $(k-1)$-manifold $X$.

the curvature is the anomaly polynomial $\Phi_{d+2}$, which is a particular closed $(d+2)$-form that will play a key role in what follows.

Equivalent to this picture, if we fix also a choice of background gauge field $A$, then $Z_{\mathcal{T}}[X;A]$ is no longer a simple $\mathbb{C}$-number, but must be regarded as a vector in a one-dimensional Hilbert space $\mathcal{H}^*(X;A)$ because of the ambiguity in defining its phase:

$$Z_{\mathcal{T}}[X;A] \in \mathcal{H}^*(X;A) \tag{2.1}$$

This fact, that the putative partition function of the $d$-dimensional anomalous theory is a vector in a Hilbert space, equivalently a section of a line bundle, means that $Z_{\mathcal{T}}[X;A]$ can be regarded as a *state* in a well-defined quantum field theory in *one-dimension higher*. We call this the *anomaly theory*, denoted $\mathcal{I}_{d+1}$. The fact that the Hilbert space is one-dimensional means the field theory is furthermore *invertible*, which in turn means the partition function of the anomaly theory itself, when evaluated on a $d+1$-manifold, is a pure-phase.

One can then attempt to define the partition function for the UV theory by splitting it into a modulus and an argument, where the latter is expressed via the anomaly theory. To do so, we suppose there is a $(d+1)$-manifold $Y$ whose boundary $\partial Y = X$ and to which all structures, including the gauge field $A$, smoothly extend.[3] Then we define

$$Z_{\mathcal{T}}[X;A] = |Z_{\mathcal{T}}[X;A]| \cdot Z_{\mathcal{I}}[Y;A], \qquad \text{where } \partial Y = X, \tag{2.2}$$

where we use the same symbol $A$ to denote the extension of the original gauge field to $Y$. In the case of chiral fermion anomalies the invertible theory is the exponential of the Atiyah–Patodi–Singer (APS) $\eta$-invariant [34–36], which we return to shortly in §II B.

If the formula (2.2) depends on the choice of bulk extension $Y$, then the putative quantum field theory that we attempted to define by $\mathcal{T}$ violates locality. So, to enforce locality is to demand that the anomaly theory evaluated for any two choices of extension $Y$ and $Y'$ must agree, *i.e.*

$$1 = \frac{Z_{\mathcal{I}}[Y']}{Z_{\mathcal{I}}[Y]} = Z_{\mathcal{I}}[Y_{\text{tot}}], \qquad Y_{\text{tot}} := Y' \cup_X \bar{Y} \tag{2.3}$$

where $\overline{Y}$ denotes the orientation reversal of $Y$, assuming for simplicity that every manifold is equipped with an orientation, and the operation $\cup_X$ denotes that $Y'$ and $\overline{Y}$ are glued along their equal but oppositely-oriented boundary $X$, the result of which is a closed $(d+1)$-manifold $Y_{\text{tot}} := Y' \cup_X \overline{Y}$. The first equality in (2.3) is our locality condition, while the second equality follows from the cutting and gluing property of invertible field theories [20]. In the case of chiral fermion anomalies that is our main interest, this cutting and gluing property is part of what has come to be known as the 'Dai–Freed theorem' [37].

---

[3] It is possible that no such extension exists. In $d = 4$ this can occur even when there is no gauge group; for instance, choosing $X$ to be the K3 surface, the spin structure thereon cannot be extended to any bulk 5-manifold. When a gauge group is included, say $G = SU(n)$, instanton configurations also pose an obstruction. The possibility for such 'non-nullbordant' spacetimes, which is probed by $\Omega_d(\cdot)$, does not give rise to any further anomalies, but rather to ambiguities in the partition function [20, 31–33]. This requires a choice of theta-angles be made for each generator in $\Omega_d(\cdot)$, including discrete factors.

The locality condition (2.3) is clearly satisfied if the anomaly theory $Z_{\mathcal{I}}[Y_{\text{tot}}] = 1$ on *all* closed $(d+1)$-manifolds $Y_{\text{tot}}$ with $G$-bundles and other appropriate structures, which is equivalent to the anomaly theory being the trivial theory. It has been argued that $Z_{\mathcal{I}}[Y_{\text{tot}}] = 1$ should indeed be taken as a fully consistent criterion for locality, that guarantees the theory is well-defined on all manifolds equipped with the structures taken to define the theory. This criterion moreover *implies* all (local and global) anomalies vanish in the more traditional sense, because $\{Y_{\text{tot}}\}$ includes as a subset all manifolds of the 'mapping torus' form $X \times S^1$, obtained from an initial theory on $(X, A, \dots)$ by interpolating along the $S^1$ direction to any would-be-gauge-equivalent field configuration also on $X$. See *e.g.* the discussion in [38] for a more pedestrian account of this perspective.

## B.   Chiral fermion anomalies

This work concerns the hydrodynamic limits of theories containing chiral fermions. As proven in Ref. [33], the invertible theory $Z_{\mathcal{I}}[Y]$ that precisely determines the phase of the fermion partition function, and thence governs all anomalies, is the exponentiated APS $\eta$-invariant [34–36] associated to a Dirac operator $i\slashed{D}$ extended to the bulk $Y$ with APS boundary conditions,[4] that is obtained from the kinetic term of the UV fermions. The anomaly theory is

$$Z_{\mathcal{I}}[Y; A] = \exp\left(-2\pi i \eta_{\mathbf{r}}(Y, A)\right), \tag{2.4}$$

where the $\eta$-invariant $\eta_{\mathbf{r}}(Y, A)$ is a sum of the signs of the eigenvalues $\lambda_k$ of the Dirac operator $i\slashed{D}$ on $Y$ evaluated for the representation $\mathbf{r}$ of $G$, which must be regularised *e.g.* via heat-kernel regularisation:

$$\eta_{\mathbf{r}}(Y, A) = \lim_{\epsilon \to 0^+} \sum_k e^{-\epsilon|\lambda_k|} \text{sign}(\lambda_k)/2. \tag{2.5}$$

where $k$ labels the eigenvalues. This invertible field theory carries complete non-perturbative information about the chiral fermion anomaly [33].

As discussed in the previous section, the phase by which the partition function transforms is expressed as the anomaly theory $Z_{\mathcal{I}}$ evaluated on some $Y_{\text{tot}}$, which is a closed $(d+1)$-manifold equipped with a spin structure and a $G$-bundle such as a mapping torus or a mapping sphere. If this can be extended to a $(d+2)$−dimensional manifold $W$ whose boundary is $\partial W = Y_{\text{tot}}$, then one can use the APS index theorem to write the exponentiated $\eta$-invariant in terms of the index density *a.k.a.* the anomaly polynomial $\Phi_{d+2}$ that captures all perturbative anomalies,

$$\exp(-2\pi i \eta_{\mathbf{r}}(Y_{\text{tot}}, A)) = \exp\left(-2\pi i \int_W \Phi_{d+2}(A, \mathbf{r})\right), \quad \Phi_{d+2}(A, \mathbf{r}) = \hat{A}(R)\, \text{tr}_{\mathbf{r}}\left[\exp\left(\frac{F}{2\pi}\right)\right], \tag{2.6}$$

---

[4] We do not go into the technicalities of defining APS boundary conditions here, but refer readers to [33].

where $\hat{A}(R)$ denotes the Dirac genus. Note that the contribution to the APS index theorem from the index itself vanishes upon the taking the complex exponential.

Such an extension may not exist when the spin bordism group, which measures obstructions to extending $Y_{\rm tot}$ to $W$, is non-trivial *i.e.* $\Omega_{d+1}^{\rm Spin}(BG) \neq 0$. In this scenario, the exponentiated $\eta$-invariant and hence the anomaly may still be non-trivial even when the index density $\Phi_{d+2} = 0$ *i.e.* when perturbative anomalies vanish. In this case that $\Phi_{d+2} = 0$, the APS index theorem tells us further useful information, namely that the exponentiated $\eta$-invariant is trivial on all manifolds that *can* be extended, which together form the zero element in the bordism group. Equivalently the anomaly, which is necessarily a *global* or *non-perturbative* one now, becomes a *bordism invariant* under these conditions, and so is a homomorphism from $\mathrm{Tor}\,\Omega_{d+1}^{\rm Spin}(BG)$ to $U(1)$.[5] An immediate corollary is that there can be no global anomalies if $\mathrm{Tor}\,\Omega_{d+1}^{\rm Spin}(BG) = 0$. More generally, when $\mathrm{Tor}\,\Omega_{d+1}^{\rm Spin}(BG)$ is not zero this means that we only need to compute the $\eta$-invariant for $Y_{\rm tot}$ being a finite-order generator of the bordism group, and once we have done so for all such generators we will have obtained complete information regarding the anomaly. For instance, $\mathrm{Tor}\,\Omega_5^{\rm Spin}(BSU(2)) \cong \mathbb{Z}_2$, meaning there is a class of spin-manifolds equipped with $SU(2)$-bundles that cannot be realised as boundaries, and we need to evaluate the $\eta$-invariant on one such manifold to compute the $SU(2)$ anomaly. We return to this example, adapted to the hydrodynamic limit, in §IV.

## III. NON-PERTURBATIVE ANOMALY MATCHING IN THERMAL EQUILIBRIUM

We wish to apply this general formalism for analysing global anomalies in the hydrodynamic, high temperature limit, assuming that the UV theory is gapped when placed on the background geometry with thermal cycle.[6] This limit involves putting the system at finite temperature, coarse-graining over all microscopic degrees of freedom, and imposing an equilibrium condition which means the background metric admits a time-like Killing vector $\partial_t$. The hydrodynamic theory is assumed to be in the *normal phase*, which means the conserved currents such as the stress-energy tensor $T_{\mu\nu}$ are expressed as functions of thermodynamic variables such as temperature $T$ and fluid velocity $u^\mu$, as well as the background fields like the metric and the background gauge field $A$ for symmetry $G$.

We suppose the microscopic theory has global symmetry $G$, and consider coupling to a background gauge field $A$ as before that transforms as

$$A \to g^{-1}Ag + g^{-1}dg \,, \qquad g = g(x^\mu) : X \to G \,. \tag{3.1}$$

---

[5] The restriction to the torsion subgroup is because any non-trivial $\eta$-invariant associated to free factors in $\Omega_{d+1}$ can be cancelled by an extra term in the partition function of the form $Z_{\rm ct} = \exp\left(-i \int \varphi\right)$ [33], where $\varphi$ is a gauge-invariant differential form constructed from $A$ and possibly the metric. This is simply some characteristic class, which can only be non-trivial when $d + 1$ is even *i.e.* when $d$ is odd.

[6] For microscopic theories of massless chiral fermions, anomalies can in fact present an obstruction to this hypothesis, at least in the presence of instantons. We comment on this later.

To go to the high temperature phase of this theory, we take the specific spacetime topology

$$X = S^1_\beta \times \mathcal{M} \,, \tag{3.2}$$

equipped with a metric and gauge field that can be cast in the Kaluza–Klein form [29],

$$g_{\mu\nu} dx^\mu dx^\nu = e^{2\sigma(x)} (d\tau + \alpha_i dx^i)^2 + \gamma_{ij} dx^i dx^j \,, \quad A = -\mu u + \mathcal{A} \,, \quad \text{where} \;\; u^\mu u_\mu = -1 \,. \tag{3.3}$$

Here $u^\mu$ plays the role of the fluid velocity such that $u = u_\mu dx^\mu = -(d\tau + \alpha_i dx^i)$, $\mu := u^\mu A_\mu$ plays the role of the chemical potential valued in the adjoint representation, and $\mathcal{A} = A + \mu u = \mathcal{A}_i dx^i$ only contains 'spatial' components orthogonal to $u^\mu$. The temperature is encoded through $1/T = \beta e^\sigma$ where $\beta$ is the size of the thermal cycle. We then dimensionally reduce the theory on the thermal cycle $S^1_\beta$, with $\beta$ assumed to be small compared to gradients of the hydrodynamic variables at high temperature.

The resulting thermal partition function $Z[g, A]$ is not invariant under all transformations of type (3.1), but only under background gauge transformations of the form

$$A_\tau \to h^{-1} A_\tau h \,, \qquad A_i \to h^{-1} A_i h + h^{-1} d_i h \,, \qquad h = h(x^i) : \mathcal{M} \to G \,, \tag{3.4}$$

due to the hypothesis that the system is in equilibrium [29]. The absence of the time-dependence in the transformation parameters signifies the breaking of boosts, and has been exploited extensively in building EFTs for hydrodynamics [39, 40] (see also [41, 42] for discussions from the correlation function perspective). Note that the 'image' of the unbroken gauge transformations remains the full group $G$, so this should be viewed purely as the breaking of a spacetime symmetry.[7]

The equilibrium partition function $Z_\mathcal{T}[g, A]$ is defined, according to [29, 46], to be a trace of a thermal density matrix, with the possible insertion of sources. One could think of it as the partition function on the background $X$ with timelike Killing vector $u^\mu/T$. The assumption that the theory on $S^1 \times \mathcal{M}$ is gapped indicates that one can integrate out all microscopic degrees of freedom and express the effective action in terms of the variables in (3.3), as

$$-\log Z_\mathcal{T}[g, A] = \beta \int_\mathcal{M} d^3x \sqrt{\gamma} \, \mathcal{L}_{\text{eff}}[\sigma, \alpha_i, \gamma_{ij}, \mu, \mathcal{A}_i] \,, \tag{3.5}$$

If the theory is totally invariant under all diffeomorphisms and global symmetries described by $h : \mathcal{M} \to G$ in (3.4), then the Lagrangian at zeroth derivative level can only depend on the temperature (through $\sigma$) and chemical potential $\mu$ that produce the ideal fluid constitutive relations for $T^{\mu\nu}$ and $J^\mu$.

When the system is anomalous, one must allow for non-invariant combinations of hydrodynamic variables (3.3) in such a way that the partition function of the theory on $(X, A)$

---

[7] This statement can also be derived using holography. Such a holographic approach has been used to extract hydrodynamics *e.g.* for a theory with $U(1)$ symmetry [43], for anomalous and non-abelian symmetry [44], and for 1-form $U(1)$ and toric 2-group symmetry [45].

and some transformed $(X', A')$ obtained by doing a background gauge transformation and possible diffeomorphism satisfies the relation dictated by the anomaly:

$$\frac{Z_{\mathcal{T}}[X'; A']}{Z_{\mathcal{T}}[X; A]} = Z_{\mathcal{I}}[Y_{\text{tot}}], \tag{3.6}$$

where $Y_{\text{tot}}$ is a mapping torus that computes the anomaly in the fundamental theory, as described in §II. This approach has been successfully applied to put constraints on the anomalous transport coefficients which control specific non-invariant combinations in systems with perturbative anomalies [29, 47–49], where $A$ and $A'$ are related by infinitesimal gauge transformations. The case of global anomalies, where the index density $\Phi_{d+2} = 0$, is more subtle as one would have to find specific mapping tori that not only probe the anomaly but are also compatible with thermal equilibrium, as advertized in step (ii) in the previous §II. This has been done in a handful of examples [24, 27, 28].

In the case of chiral fermion anomalies, some further general comments are warranted before we turn to the $USp(2N)$ case:

- Our assumption that the UV theory is gapped, necessary to go to the hydrodynamic limit, prevents us from considering instanton configurations for the background gauge field $A$ on $X$, since non-zero instanton number would imply by the Atiyah–Singer index theorem that there are fermion zero modes.

- Zero gauge instanton number implies that $[X] = 0 \in \Omega_4^{\text{Spin}}(BG)$, in the case of interest where $G$ is a simple Lie group, provided we choose a spin structure that can be filled in. This means the assumption that $X$ can be extended to a bounding $Y$ holds in our context, and the ambiguities associated with assigning the partition function on non-nullbordant $X$ are not activated. In other words, the generalised theta-angles for the gauge symmetry are in fact not physical in our hydrodynamical EFT limit.

- The reader might already be puzzled, given the previous two points, how we can hope to derive consequences of *e.g.* the $SU(2)$ global anomaly when we forbid instanton configurations on $X$. The point will be, however, that one can still probe mapping tori that are in the non-trivial 5[th] bordism group, for instance by wrapping an instanton around the product of the spatial $\mathcal{M}$ with the auxiliary mapping torus direction (rather than with the time-like coordinate in $X$). But, as we will see, this also requires a non-trivial discrete holonomy wrapping the thermal cycle, which we will interpret as a defect required to detect the presence of a global anomaly.

## IV. MATCHING THE WITTEN ANOMALY IN THERMAL EQUILIBRIUM

Recall that global anomalies, which are discrete phase ambiguities in the partition function, are determined by the bordism group $\text{Tor}\,\Omega_5^{\text{Spin}}(BG)$. The spin bordism groups

$\Omega_5^{\mathrm{Spin}}(BG)$ vanish for all $G = SU(N)$ with $N \geq 3$, for all $G = SO(N)$, and for all the exceptional Lie groups *i.e.* $G_2$, $F_4$, and $E_{6,7,8}$ [50]. The only simple Lie groups with non-trivial degree-5 bordism groups and thus global anomalies are of course

$$\Omega_5^{\mathrm{Spin}}(BUSp(2N)) \cong \mathbb{Z}_2 \qquad \forall N \geq 1\,, \tag{4.1}$$

including the special case $USp(2) \cong SU(2)$. Being $\mathbb{Z}_2$-valued implies the most general anomaly would be a sign flip of the partition function,

$$Z[A] \rightarrow Z[A^g] = -Z[A]\,. \tag{4.2}$$

The anomaly is indeed present for all $USp(2N)$ including $SU(2)$.

Taking $G = SU(2)$ for concreteness and simplicity (but noting the same holds for any $USp(2N)$), there are two standard mapping torus constructions in the literature that lie in the non-trivial bordism class and that probe the global anomaly. The first is that originally described by Witten in Ref. [13], in which the Euclideanised spacetime is assumed to be a 4-sphere, and a map $g(x) : S^4 \rightarrow SU(2)$ in the non-trivial homotopy class is used to glue together the ends of a mapping torus, starting from any $SU(2)$ gauge connection on $S^4$. The second is described more recently by Wang, Wen and Witten in Ref. [14], whereby spacetime $S^4$ is equipped with an $SU(2)$ background with odd instanton number, and the constant gauge transformation $g(x) = -1$ is used to glue the ends of the mapping torus.

Neither of these descriptions are consistent with our hydrodynamic setup in which, for instance, the 4-d Euclidean spacetime is required to be a circle fibration over a spatial 3-manifold, and has no instanton (see §III). In this section we nonetheless show how $SU(2)$ gauge transformations can probe the $SU(2)$ global anomaly on spacetime backgrounds that are consistent with our hydrodynamic hypothesis (§IV A). The picture generalises with minor modifications to any $USp(2N)$ (§IV B).

## A. Clutching construction for SU(2) anomalies in hydrodynamics

We consider a theory that in the UV has chiral fermions transforming in representation $\mathbf{r}$ of a global $SU(2)$ symmetry. For simplicity, let us assume that, at thermal equilibrium, the underlying manifold is a trivial circle fibration with the spatial manifold being a 3-sphere,

$$X_4 = S_\beta^1 \times S^3\,. \tag{4.3}$$

Compatibility with thermal equilibrium requires that any background gauge field on $X_4$, as well as any gauge transformation $g$, be independent of the thermal cycle direction. The partition function receives a contribution $Z_\mathcal{I}[Y, A] = \exp(-2\pi i \eta_\mathbf{r}(Y, A))$ from the exponentiated $\eta$-invariant of a 5-d Dirac operator acting on our $SU(2)$ representation, where we extend spacetime $X_4$ as the boundary of a 5-manifold

$$Y = S_\beta^1 \times D^4, \qquad \partial D^4 = S^3 \tag{4.4}$$

with appropriate boundary conditions on the Dirac operator. Such an extension of the spatial $S^3$ to a disc is always possible because

$$\pi_3(BSU(2)) \cong \pi_2(SU(2)) = 0\,. \tag{4.5}$$

As explained above, compatibility with equilibrium requires that any gauge transformation $g$, which is in general a map from $X_4 = S_\beta^1 \times S^3$ into $SU(2)$, should be independent of $S_\beta^1$. Thus $g$ reduces to a map from the spatial manifold only, *i.e.*

$$g(x^i) : S^3 \to SU(2)\,, \tag{4.6}$$

and so effectively implements a 3-d gauge transformation. Such maps are classified by their homotopy class, which here coincides with the 'degree' $n$ of a map from $S^3$ to itself,

$$[g] = n \in \pi_3(SU(2)) \cong \mathbb{Z}\,. \tag{4.7}$$

Under such a gauge transformation $A \mapsto A^g$, we track the variation of the partition function phase through the invertible theory, which transforms from $Z_\mathcal{I}[S_\beta^1 \times D^4; A]$ to $Z_\mathcal{I}[S_\beta^1 \times D^4; A^g]$, where the $SU(2)$-bundle over each $D^4$ is trivial because $D^4$ is contractible.

The anomalous phase variation can then be computed via

$$\frac{Z_\mathcal{I}[S_\beta^1 \times D^4; A^g]}{Z_\mathcal{I}[S_\beta^1 \times D^4; A]} = Z_\mathcal{I}[(S_\beta^1 \times D^4; A^g) \cup_X (S_\beta^1 \times \overline{D^4}; A)] \tag{4.8}$$

$$= Z_\mathcal{I}[S_\beta^1 \times S^4; A_{\text{clutch}}] = \exp\left(-2\pi i \eta_{\mathbf{r}}\left(S_\beta^1 \times S^4, A_{\text{clutch}}\right)\right)$$

where in the first equality we reverse orientation to move the factor in the denominator to the numerator. The second equality is the crucial step: the $S^4$ is obtained by gluing the two $D^4$ hemispheres together along their shared boundary $X_3$, with the gauge fields $A$ and $A^g$ glued together with the gauge transformation $g$ at the equator $S^3$, as shown in Fig. 1. The $S_\beta^1$ factor is essentially a spectator to this procedure.

Gluing two hemispheres (here $D^4$) with $G$-bundle in this way to form a sphere (here $S^4$), via a non-trivial gauge transformation $g : S^3 \to G$ on the equator, is known in topology as the *clutching construction*, with $g$ being the clutching function. The resulting $SU(2)$-bundle on $S^4$, with connection $A_{\text{clutch}}$, is specified by the degree $n$ of the clutching function $g : S^3 \to SU(2)$ used to glue the two 4-discs, which recall started life as an effective 3-d gauge transformation consistent with our assumptions of thermal equilbrium. Precisely, the integer $n$ becomes the instanton number of the bundle over the $S^4$, *viz.*

$$\langle [S^4], p_1(F_{\text{clutch}})\rangle = \deg(g) = n\,, \tag{4.9}$$

where $p_1(F_{\text{clutch}})$ is the first Pontryagin class of $A_{\text{clutch}}$. This identification can also be understood from the long exact sequence in homotopy applied to $G \to EG \to BG$, using the fact that $EG$ is contractible, which yields the isomorphism $\pi_3(SU(2)) \cong \pi_4(BSU(2))$.

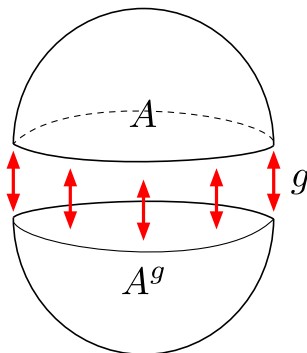

FIG. 1: The 'clutching construction' of a $SU(2)$-bundle on $S^1 \times S^4 = S^1_\beta \times (D_4 \cup_{X_3} \bar{D}_4)$. Here we suppress the $S^1_\beta$ direction, and represent the manifolds $D^4$ and $S^4$ as 2-discs and 2-spheres, respectively. The resulting $SU(2)$ bundle on $S^4$ has instanton number equal to the homotopy class in $\pi_3(S^3)$ of the 3-d gauge transformation $g(x^i)$ used in the gluing.

Thus, the anomalous phase variation is computed by evaluating the exponentiated $\eta$-invariant on a product of the thermal cycle with an $S^4$ equipped with an $n$-instanton. In other words, we arrive at the same manifold with $G$-bundle as that formulated by Wang, Wen, and Witten in [14]. While there it was formed as a mapping torus, here it is formed as a 'mapping sphere', with the $S^1$ direction not parametrizing the gauge transformation but rather being the thermal cycle $S^1_\beta$ in the original $X^4$. For this 5-manifold to be in the non-trivial bordism class, we know that we need to arrange for the non-trivial (i.e. periodic) spin structure around $S^1_\beta$.

The APS $\eta$-invariant on this manifold is well-known, and can be arrived at through a variety of methods, for instance via a mod 2 index theorem [51], or via the more pedestrian method of 'anomaly interplay' [52–54] that we review in Appendix A. Perhaps the most direct way is to use the following 'factorization' property that holds for the Dirac operator under consideration [55, Lemma 2.2]

$$\exp 2\pi i \left[ \eta(S^1_\beta \times S^4) \right] = \exp 2\pi i \left[ \eta(S^1_\beta) \times \text{Ind}(S^4) \right] , \qquad (4.10)$$

as described for example in [50, 56], and recently in Ref. [57] in the context of the 3+1d $SU(2)$ anomaly we consider here. The $\eta$-invariant on $S^1_\beta$ is determined by the spin structure of the fermion. For the AP spin structure, there is no zero mode on $S^1_\beta$ and thus $2\eta(S^1) = 0$ mod 2. For the P structure, there is a zero mode and direct computation using (2.5) results in $2\eta = 1$ mod 2, which verifies that $S^1_\beta$ with P structure is non-trivial in spin bordism. This implies

$$\exp\left(-2\pi i \eta_{\mathbf{r}}\left(S^1_\beta \times S^4, A_{\text{clutch}}\right)\right) = \begin{cases} 1, & S^1_\beta \text{ has AP spin structure,} \\ (-1)^{\int_{S^4} \Phi_4(A_{\text{clutch}}, \mathbf{r})}, & S^1_\beta \text{ has P spin structure.} \end{cases} \qquad (4.11)$$

Here $\Phi_4$ is the degree-4 anomaly polynomial appearing in the Atiyah–Singer theorem,

$$\int_{S^4} \Phi_4(A_{\text{clutch}}, \mathbf{r}) = -\frac{1}{8\pi^2} \int_{S^4} \text{Tr}_{\mathbf{r}}(F^2_{\text{clutch}}) = -T(j) \int_{S^4} p_1(F_{\text{clutch}}), \qquad (4.12)$$

where the Dynkin index $T(j) = \frac{2}{3}j(j+1)(2j+1)$, which is defined by $\text{Tr}\ (t^a_j t^b_j) = \frac{1}{2}\delta^{ab}T(j)$, is odd for $j \in 2\mathbb{Z} + \frac{1}{2}$ and even otherwise, and $\int p_1(F_{\text{clutch}}) = n$ is the instanton number of the $SU(2)$ bundle over $S^4$ that we obtained via the clutching construction.

Thus, the thermal partition function changes sign under the $SU(2)$ gauge transformation $g : S^3 \to SU(2)$ that we started with iff:

1. The degree $n$ of the spatial gauge transformation map $g$ is odd;

2. The thermal cycle $S^1_\beta$ is equipped with periodic spin structure;

3. There is an odd number of chiral fermions transforming in representations with $j \in 2\mathbb{Z} + \frac{1}{2}$.

The final condition is of course the usual one for the fundamental theory to exhibit the $SU(2)$ anomaly.

The first two conditions constitute our main finding of how the $SU(2)$ anomalous transformation can be seen in thermal equilibrium. In particular, the second condition tells us that, when the $S^1_\beta$ has anti-periodic spin structure, the effective description in the thermal state is indistinguishable from the anomaly-free one. Thus, to see the effect of the $SU(2)$ anomaly, one needs to insert an operator into the thermal partition function that flips the fermion boundary condition from anti-periodic to periodic. For example, we may consider the twisted partition function

$$Z^{\text{twisted}}_{\text{thermal}} = \text{tr}\left[(-1)^F e^{-\beta H}\right] \quad \text{or} \quad \text{tr}\left[U_\pi e^{-\beta H}\right], \qquad (4.13)$$

where $U_\pi$ is the central element of $SU(2)$ that flips the sign of, for instance, a fundamental fermion as it passes through this operator when going around the thermal cycle. It should be emphasised that this possibility depends on the representation of the fermion microscopic constituents. For integer isospins $j \in \mathbb{Z}$, the representation is of course real and there is no unpaired zero mode on $S^1_\beta$ rendering $2\eta(S^1)$ to vanish mod 2 (in our conventions). This can also be understood from the fact that the central element $U_\pi \in SU(2)$ acts trivially on all integer $j$ representations and therefore the spin structure cannot be flipped from the AP to the P one by inserting $U_\pi$. For half-integer $j$, however, there can be zero mode upon the centre insertion and the regulated sum of eigenvalues in (2.5) yields $2\eta(S^1) = j$ mod 1.

## B. Generalisation from SU(2) to USp(2N) anomalies

Our clutching construction argument, with which we build a non-trivial 'mapping sphere' that consistently probes the $SU(2)$ anomaly in thermal equilibrium, can be generalised to the

case of global symmetry $G = USp(2N)$, which we know suffers from the same microscopic global anomaly [13]. Throughout this section we quote various homotopy groups of $USp(2N)$ up to degree 5, which were computed by Bott [58]. See also [59].

Our starting point is again a $USp(2N)$-bundle over $X_4 = S^1_\beta \times S^3$, which can again be extended as the boundary of a 5-manifold $Y = S^1_\beta \times D^4$, because

$$\pi_3(BUSp(2N)) \cong \pi_2(USp(2N)) = 0 \qquad (4.14)$$

for any $N$. We consider a purely spatial 3-d gauge transformation $g : S^3 \to USp(2N)$, consistent with the assumption of thermal equilibrium. This is now classified up to homotopy by an element

$$[g] = n \in \pi_3(USp(2N)) \cong \mathbb{Z}. \qquad (4.15)$$

Again, to compute the change in the partition function phase for $A^g$ *vs.* $A$ backgrounds, we evaluate the anomaly theory on a 5-d mapping sphere obtained by gluing together the two 4-discs with $USp(2N)$-bundle along their equator via the clutching function $g$. The clutching construction tells us that the resulting bundle on $S^4$ is classified by an element in the homotopy class of maps $[S^4, BUSp(2N)] \cong \pi_4(BUSp(2N)) \cong \mathbb{Z}$ equal to $n = [g]$. Repeating the same argument as for $SU(2)$, the anomalous phase variation is then computed by the mapping sphere, giving the same result (4.11).

The difference is only in performing the trace over the fermion representation, which is a little more involved for general $USp(2N)$. Now a representation is labelled by $N$ Dynkin labels or equivalently $N$ 'isospin' quantum numbers $j_1, \ldots, j_n$, and $T(j)$ is replaced by the Dynkin index $T(j_1, \ldots, j_n)$ which is again defined via $\text{Tr}\,(t^a_{j_1, \ldots, j_N} t^b_{j_1, \ldots, j_N}) = \frac{1}{2}\delta^{ab} T(j_1, \ldots, j_N)$. For the fundamental **2N** representation, this is again equal to unity in our convention. More generally, as for the $SU(2)$ case, the global anomaly is only probed when there is an odd number of chiral fermions in total that transform in representations for which $T(j_1, \ldots, j_n)$ is odd.

## V. CONSEQUENCES IN TRANSPORT PHENOMENA

In its simplest form, hydrodynamics is defined by a set of gradient expansions of the conserved quantities in terms of their conjugate variables. See *e.g.* [60] for a modern review. The consequences of perturbative 't Hooft anomalies in transport phenomena have been derived, for example using the positivity of entropy production [61] or by matching the variation of the thermal partition function [29, 48]. The form of the conserved current $J^\mu$ and its response due to vorticity and magnetic field are constrained by the anomaly.

## A.   Review: non-abelian perturbative anomaly-induced transport

A generalisation to non-abelian global symmetries in hydrodynamics has been carried out in the last few decades.[8] There the density and chemical potential are promoted to matrix-valued quantities that transform in the adjoint representation of a group $G$. Consider the case where the *covariant* current $J_a^i := 2\mathrm{tr}[J^i \mathsf{T}_a]$ obeys the following anomalous Ward identity

$$D_\mu J^{a\mu} = \frac{1}{8} C^{abc} \epsilon_{\mu\nu\rho\sigma} F_b^{\mu\nu} F_c^{\rho\sigma}, \qquad C^{abc} = \frac{1}{4\pi^2} \mathrm{tr}\left[ \mathsf{T}_{(a} \mathsf{T}_b \mathsf{T}_{c)} \right] \tag{5.1}$$

with $\{\mathsf{T}_a\}$ are the generators of $G$. It can be shown, by demanding the positivity of the entropy production and CPT symmetry, that the stress-energy tensor and the current must contain the following terms [70][9]

$$\begin{aligned}
T^i_{\ t} &= -\left( \frac{1}{3} C_{bcd} \mu^b \mu^c \mu^d + 2\beta_b \mu^b T^2 \right) \omega^i + \left( \frac{1}{2} C_{abc} \mu^b \mu^c + \beta_a T \right) B^{ai}, \\
J_a^i &= \left( \frac{1}{2} C_{abc} \mu^b \mu^c + \beta_a T^2 \right) \omega^i + C_{abc} \mu^b B^{ci}
\end{aligned} \tag{5.2}$$

where $\beta_a$ are constants that depend neither on the temperature nor on the chemical potential components $\mu_a := 2\mathrm{tr}[\mu \mathsf{T}_a]$. The same conclusion follows from the consistent variation of the equilibrium partition function [49].

From the perspective of the dimensionally-reduced effective action (3.5), the undetermined coefficients $\beta_a$ originate from mixed Chern–Simons term involving the Kaluza–Klein gauge field $\alpha$ defined in (3.3) and the gauge field $\mathcal{A}$ that appears at high temperature,

$$-i \log Z \supset \frac{n}{2\pi} \int_{\mathcal{M}} \alpha \wedge \mathrm{tr}[d\mathcal{A}]. \tag{5.3}$$

Observe that the other possible Chern–Simons terms involving only $\alpha$ or only $\mathcal{A}$, namely

$$\int_{\mathcal{M}} CS_3[\alpha] := \int_{\mathcal{M}} \alpha \, d\alpha, \qquad \int_{\mathcal{M}} CS_3[\mathcal{A}] := \int_{\mathcal{M}} \mathrm{tr}\left[ \mathcal{A} \wedge d\mathcal{A} + \frac{2}{3} \mathcal{A} \wedge \mathcal{A} \wedge \mathcal{A} \right] \tag{5.4}$$

with properly quantised coefficients, generically violate 4d CPT symmetry and are therefore omitted from the analysis [28, 29]. See the table below for the transformations of hydrodynamic variables under CPT.

---

[8] To our knowledge, the earliest formulation of hydrodynamics with non-abelian global symmetry used the kinetic theory framework [62] (see [63] for a review). Early macroscopic approaches can be found in [64, 65] with applications in (iso)spin transport [66]. Our approach in this section more closely follows those in [67–69], which can be derived from gauge/gravity duality as well as being supported by the results of relevant classical simulations.

[9] The result we quote follows from the analysis of [70] but without imposing the 'Landau frame' where $T^{\mu\nu} u_\nu = -\varepsilon u^\mu$ and $J_a^\mu u^\mu = -n_a$ with $\varepsilon, n_a$ being the energy and non-abelian charge density in equilibrium.

| Discrete transformations in 3+1 dimensions | | | | |
|---|---|---|---|---|
| | T | P | C | CPT |
| $x^\mu$ | $(-x^0, x^i)$ | $(x^0, -x^i)$ | $(x^0, x^i)$ | $-(x^0, x^i)$ |
| $u^\mu$ | $(u^0, -u^i)$ | $(u^0, -u^i)$ | $u^\mu$ | $u^\mu$ |
| $\alpha_i$ | $-\alpha_i$ | $-\alpha_i$ | $\alpha_i$ | $\alpha_i$ |
| $\omega^i \sim \epsilon d\alpha$ | $-\omega^i$ | $\omega^i$ | $\omega^i$ | $-\omega^i$ |
| $A_\mu$ | $(A_0, -A_i)$ | $(A_0, -A_i)$ | $-A_\mu$ | $-A_\mu$ |

## B. Fractional transport from the Witten anomaly

When the symmetry group $G$ is $SU(2)$, or more generally $USp(2N)$, the perturbative anomaly coefficients $C_{abc}$ of course vanish, and the tracelessness of the generators also implies that the terms corresponding to $\beta_a$ vanish also. At first glance, one might be tempted to conclude that there can be no effect due to the global anomaly for these groups, both at the level of the conserved currents and the equilibrium partition function. However, this would contradict the 't Hooft anomaly matching argument. Therefore, either the normal phase is not compatible with the 't Hooft anomaly, which we explicitly demonstrated is not the case in Section IV by constructing an appropriate mapping sphere, or one has to circumvent the naïve constraints coming from CPT invariance and cook up terms in $\log Z$ that can match the global anomaly.

The $SU(2)$ global anomaly is related to the mod 2 reduction of the Pontryagin class $p_1(F)$ for non-trivial $SU(2)$-bundles, and we know that $\int p_1(F)$ is in turn related to the 3d Chern–Simons term $CS_3[\mathcal{A}]$ via inflow, which suggests $CS_3[\mathcal{A}]$ could play a role in matching the anomaly. Having just argued that $CS_3[\mathcal{A}]$ is CPT-odd, the only way to include such a term in the effective action is if its quantised coefficient is also CPT-odd. Recall also from §IV A that the mod 2 anomaly is probed in the high temperature limit only if the thermal cycle $S_\beta^1$ has periodic spin structure, with the anomaly theory being proportional to $\eta(S_\beta^1)$ as in (4.10).

With this guidance, we introduce the following fractional Chern–Simons term into the effective action for our hydrodynamic theory reduced on the thermal cycle $S_\beta^1$:

$$-i\log Z \supset 2\eta(S_\beta^1)\int_{\mathcal{M}} \frac{1}{2}\frac{q(j)}{4\pi}CS_3[\mathcal{A}]\,, \qquad q(j)\in\mathbb{Z}\,. \tag{5.5}$$

The coefficient of the 3-d Chern–Simons term, proportional to $\eta(S_\beta^1)q(j)$, does indeed flip sign under CPT which sends $\eta(S_\beta^1) \to -\eta(S_\beta^1)$.[10] This CPT-odd nature of the coefficient is reminiscent of the flip $\theta \mapsto -\theta$ under charge conjugation imposed in *e.g.* [71], which is

---

[10] For the 1-d Dirac operator on $S^1$, CPT just sends $t \to -t$ and thus flips the sign of the Dirac operator, thence flips the sign of all of its eigenvalues $\lambda_k$, thence flips the sign of $\eta(S^1)$.

needed for the $\theta$−term to preserve CPT. Notice that $\int \frac{1}{4\pi} CS_3$ would be an integer level Chern–Simons term; the mod 2 fractional level is activated for P spin structure *i.e.* when $2\eta = 1 \mod 2$.

With our candidate effective action (5.5) in hand, one can then proceed to demand that it matches the anomaly, which requires

$$\exp\left(2\eta(S_\beta^1)\frac{iq(j)}{8\pi}\int_{\mathcal{M}}\left(CS[\mathcal{A}^g] - CS[\mathcal{A}]\right)\right) = \exp\left(-2\pi i \eta_j(S_\beta^1 \times S^4, A_{\text{clutch}})\right), \quad (5.6)$$

where $\mathcal{A}^g$ is related to $\mathcal{A}$ by a transition function $g$, characterized by its class $[g]$ in $\pi_3(G)$, and recall $A_{\text{clutch}}$ is the gauge field on $S^4$ obtained using $g$ as a clutching function. Evaluating the left-hand-side, namely the variation of the 3-d non-abelian Chern–Simons action, is a standard computation (see *e.g.* [72]), and gives

$$\exp\left(2\eta(S_\beta^1)\ i\pi q(j)\int_{\mathcal{M}}\frac{\text{Tr }(g^{-1}dg)^3}{24\pi^2}\right) = (-1)^{2\eta(S_\beta^1)\,q(j)[g]}. \quad (5.7)$$

Recall from §IV A that the anomalous variation $\exp 2\pi i \eta_j(S_\beta^1 \times S^4)$ is non-trivial only when the three conditions

$$[g] \in \mathbb{Z}_{\text{odd}}, \quad (5.8)$$

$$2\eta(S_\beta^1) = 1 \mod 2 \quad (\text{P spin structure}), \quad (5.9)$$

$$T(j) \in \mathbb{Z}_{\text{odd}} \quad (5.10)$$

are all satisfied. Thus, to match the anomaly we have the following condition on the coefficient of the fractional Chern–Simons term:

$$q(j) = T(j) \mod 2. \quad (5.11)$$

Equivalently, $q(j)$ must be odd for $j \in 2\mathbb{Z} + \frac{1}{2}$, and even otherwise. We emphasize that the integrality of $q(j)$, which means the fractional Chern–Simons term (5.5) evaluates only to $\pm 1$ depending on the choice of background and of representation, means it is invariant under CPT.

The generalisation to other $USp(2N)$ is straightforward. Now the coefficient of the fractional Chern–Simons term $q(j_1, ..., j_N)$ must be an odd integer iff the fermion transforms in a representation $|j_1, ..., j_N\rangle$ whose $\mathbb{Z}_2$ central element acts nontrivially and that the Dynkin index $T(j_1, ..., j_N)$ is odd.

To see how this fractional Chern–Simons term in the effective action affects the conserved currents, we consider the variation with respect to $\alpha_i$ and $\mathcal{A}_i$ in the Kaluza–Klein coordinate, similar to the variations considered in [29]. From this, we conclude that the *consistent* conserved currents $T^t_{\ t}$ and $J^i_{\ a}$ in the anomalous theory must contains the terms

$$\begin{aligned} T^i_{\ t} &= \frac{q(j)}{8\pi}\mu^a T B^i_a + \frac{q(j)}{8\pi}\mu^a\mu_a T\omega^i, \\ J^i_a &= \frac{q(j)}{8\pi}T B^i_a + \frac{q(j)}{8\pi}\mu_a T\omega^i, \end{aligned} \quad (5.12)$$

which are not present in the corresponding analysis of fluids with perturbative 't Hooft anomalies, as in [70]. Thus, we derive new transport effects akin to the chiral magnetic effect and its cousins, that arise not due to a perturbative anomaly but due to a more subtle global anomaly.

It should be emphasised that the global anomaly matching only determines the value of the transport coefficient $q$ modulo 2, *i.e.* whether it is an even or odd integer. For example, in the case of a fluid with $SU(2)$ global symmetry for which the microscopic degree of freedom is a massless fermion in the fundamental $j = 1/2$ representation with, we require

$$q(j) = 2k + 1, \qquad k \in \mathbb{Z} \tag{5.13}$$

The integer $k$ is not fixed by the anomaly, but rather is determined by the microscopic details. The fractional part of $q(j)/2$ is fixed by the 't Hooft anomaly matching. This 'fractionally-induced transport' mirrors that derived from global anomalies involving $\mathbb{Z}_2$ global symmetries in [27, 28]. See [73] for similar arguments related to 3-d QFTs at zero temperature.

Before concluding, we point out a few subtleties. First, note that the currents in (5.2) and (5.12) are of different nature: the former is known as the *covariant* current,[11] obtained by attaching the anomalous theory on $X = \partial Y$ to a bulk Chern–Simons theory on $Y$ that cancels the anomaly via inflow, whereas the currents we use in (5.12) are derived from the anomalous partition function with no bulk attached. Moreover, we emphasize that the thermal partition function that would result in (5.12) requires the P spin structure on the compactified thermal cycle, which can be implemented by 'twisting':

$$Z_{\text{thermal}}^{\text{twisted}} = \text{tr}\left[(-1)^F e^{-\beta H}\right] \quad \text{or} \quad \text{tr}\left[U_\pi e^{-\beta H}\right].$$

Had there been no twist, it follows that the partition function is invariant and there is no anomaly induced current as one would have anticipated from the result of [70].

## VI. DISCUSSION AND FUTURE DIRECTIONS

In this paper we have shown that the Witten anomaly associated with $SU(2)$ global symmetry can be detected in the hydrodynamic phase at high temperature through the momentum and current response to the vorticity $\omega^i$ and applied external (non-abelian) magnetic field $B_a^i$. This phenomenon can be captured via the relations

$$\begin{pmatrix} J_a^i \\ T_t^i \end{pmatrix} = \begin{pmatrix} \sigma_{ab}^{(BB)} & \sigma_a^{(B\omega)} \\ \sigma_b^{(\omega B)} & \sigma^{(\omega\omega)} \end{pmatrix} \begin{pmatrix} B_b^i \\ \omega^i \end{pmatrix} \tag{6.1a}$$

---

[11] Here, we use the terminology of covariant and *consistent* partition functions, following [74]. A modern review of these concepts can be found in *e.g.* [48]. Curiously, the conserved currents (5.12) can also be obtained from covariant currents for a theory with perturbative $U(2)$ anomaly [54] via (5.2), for which the anomaly coefficients $C_{abc}$ are zero except for the component $C_{1bc}$ which we take to equal $q(j)$ (where $a = 1$ labels the $U(1)$ generator by which we extend $SU(2)$ to get $U(2)$), provided one also restricts the value of $J_1^\mu = 0, \mu^1 = \pi T$ and $B^{1i} = 0$.

with $a, b$ the Lie algebra indices, where the anomalous conductivities take the following non-zero values when the partition function is twisted by $\mathbb{Z}_2$:

$$\sigma^{(BB)}_{ab} = \frac{q(j)}{4\pi} T \delta_{ab} \,, \qquad \sigma^{(B\omega)}_a = \sigma^{(\omega B)}_a = \frac{q(j)}{4\pi} \mu_a T \,, \qquad \sigma^{(\omega\omega)} = \frac{q(j)}{4\pi} (\mu^a \mu_a) T \,, \qquad \text{(6.1b)}$$

where anomaly matching dictates that

$$q(j) = T(j) \mod 2 \qquad \text{(6.1c)}$$

in the $SU(2)$ case, given a fundamental theory with a microscopic Weyl fermion transforming in the isospin $j$ representation. From the perspective of the effective action, these relations follow from a fractional Chern–Simons term that is activated when the fundamental 4-d fermions are compactified on the thermal cycle $S^1_\beta$ with a periodic spin structure (which corresponds to the aforementioned twist of the partition function).

It would be valuable to confirm these results via microscopic computations, which could be obtained by extrapolating the free fermion limit (as done for a global anomaly in [28]). An important future direction is to explore how this fractional anomalous transport can be tested experimentally, as in *e.g.* [11]. One class of promising systems for probing these phenomena include multi-Weyl semi-metals which have an emergent $SU(2)$ symmetry, albeit with a different anomaly structure [75].

Our analysis of the hydrodynamic consequences of the Witten anomaly started with a formulation of an appropriate mapping torus (or 'mapping sphere') that is in the non-trivial bordism class that is furthermore consistent with passing to the hydrodynamic, high temperature limit with the assumption of being in thermal equilibrium. A simple clutching construction was used to build this bundle and compute the anomaly theory thereon. We performed a computation that was similar in spirit for a theory with $U(1) \times \mathbb{Z}_2$ global symmetry, that features a $\mathbb{Z}_4$-valued global anomaly, in [28]. In future work it would be interesting to ask whether there are global anomalies for which this general strategy would be expected to fail. For example, consider a theory in which the spacetime and internal symmetries are intertwined to form a so-called $\mathrm{Spin}_G(4)$ structure

$$\mathrm{Spin}_G(4) \cong \frac{\mathrm{Spin}(4) \times G}{\Gamma}$$

with the $\Gamma \cong \mathbb{Z}_2$ quotient identifying $(-1)^F \in \mathrm{Spin}(4)$ with a central subgroup in $G$. This type of symmetry can furnish 't Hooft anomalies [14, 76–78], for which the bordism generator – examples of which include the 'Wu manifold' $SU(3)/SO(3)$ or the Dold manifold $\mathbb{CP}^2 \rtimes S^1$ – cannot be naïvely related to a thermodynamic configuration.

A second obstruction to our analysis would occur when the global anomaly cannot be saturated by the hydrodynamic degrees of freedom; this may be the case when the anomaly is captured by domain walls, such as for those anomalies in [79].

A final obstruction to our formalism, and arguably the most interesting, occurs when our assumption that the QFT at high temperature is in the normal phase is violated. There

is a growing class of models that are known to exhibit spontaneous symmetry breaking that persists at high temperature due to an anomaly involving a 1-form global symmetry [80–85]. This would contradict our assumption that the high temperature system is in the normal fluid phase. These scenarios do not seem to be mutually exclusive and it would be an interesting direction to investigate them further.

Going further in this direction, it is known that gauging a non-anomalous subgroup of an anomalous theory is one pathway to obtaining a more intricate generalised symmetry structure, such as higher-group symmetry or non-invertible symmetry [86] (see also [87–89] for reviews). Naturally, one could wonder whether the macroscopic EFTs relevant for thermodynamics and hydrodynamics can probe these rich symmetry structures. So far, the answer seems to be affirmative but the examples are limited to symmetry structures obtained by gauging subgroups of symmetries afflicted by perturbative anomalies [45, 90]. This is largely because hydrodynamic EFTs are typically formulated in terms of local currents, while global anomalies are not seen with this language. There are related constructions at zero temperature in systems of anyons [91] and for those with the global symmetry of the Standard Model [92] where fractionally quantised transport is observed, and it would be interesting to understand the response of such systems at finite temperature. We hope that the methods presented here might provide a small step forward in this direction.

## ACKNOWLEDGEMENT

We thank Mohamed Anber, Oleg Evnin, Elias Kiritsis, Eric Poppitz, Fran Peña-Benítez and Kazuya Yonekura for discussion. NP would like to thanks Asia Pacific Center for Theoretical Physics (APCTP) and Institute for Basic Science (PCS IBS) for their hospitality. The work of NP is supported by PMU-B grant number B39G670016 and Fundmental Fund grant number INDFF683692300097.

## Appendix A: Evaluating the $\eta$-invariant via anomaly interplay

In this Appendix, we briefly summarise the technique we call 'anomaly interplay' for evaluating $\eta$-invariants that probe global anomalies. Our computations will amount to calculating the exponentiated $\eta$-invariant on the mapping sphere of §II.

The idea is to relate a global anomaly associated with a unitary symmetry group $G$ to a perturbative anomaly of another group $G' \supset G$, where $G$ has no perturbative anomalies ($\Phi_{d+2} = 0$) and $G'$ has no global anomalies. This allows us to relate the global anomaly to the integral of a closed differential form, as in (2.6). The idea goes back to Ref. [5], wherein Witten showed that the mod 2 global anomaly in $SU(2)$ can be obtained from perturbative anomalies by embedding $SU(2)$ in $SU(3)$. It was further expanded upon by Elitzur and Nair in Ref. [52], albeit couched in the language of homotopy groups. This notion was

reformulated in [93, 94] in terms of cobordism in accordance with the modern classification of anomalies, which we review here.

The global anomaly of interest is detected by the exponentiated $\eta$-invariant being nontrivial on a (representative of a) non-trivial generator $(Y_{\text{tot}}, A)$ of the bordism group $\Omega_5^{\text{Spin}}(BG)$. This means one cannot extend $(Y_{\text{tot}}, A)$ as a boundary of a 6-manifold $W$ with all the structures extended, and so one cannot use for instance the APS index theorem to simplify the $\eta$-invariant expression further.

To explain how the anomaly interplay strategy provides a way forward in this situation, we first take a closer look at properties of the $\eta$-invariant. The $\eta$-invariant $\eta_{\mathbf{r}}$, computed for a Dirac operator $i\slashed{D}$ of fermions in a particular representation $\mathbf{r}$ of the symmetry group $G$, defines a map from the space of spin manifolds equipped with a $G$-bundle, which we denote by $\mathcal{M}(G)$, to $\mathbb{R}$, *i.e.*, it is an element of $\text{Hom}(\mathcal{M}(G), \mathbb{R})$. Going a step further, we can see that $\eta$ maps a representation of $G$ to an element of $\text{Hom}(\mathcal{M}(G), \mathbb{R})$:

$$\eta : RU(G) \rightarrow \text{Hom}(\mathcal{M}(G), \mathbb{R}) :$$
$$\mathbf{r} \mapsto \eta_{\mathbf{r}}, \tag{A1}$$

where $RU(G)$ is the unitary representation ring of $G$. For our purpose, we forget the multiplicative structure on $RU(G)$ and regard it as a free abelian group generated by (isomorphism classes of) irreducible representations (irreps) of $G$. Formally, an element of $\mathbf{r} \in RU(G)$ can be written as

$$\mathbf{r} = \bigoplus_i a_i \mathbf{r}_i, \qquad a_i \in \mathbb{Z}, \tag{A2}$$

where the sum is over all irreps $\mathbf{r}_i$ of $G$. When $a_i > 0$, this is the usual direct sum. When $a_i < 0$, we define it formally by saying that $(-\mathbf{r}_i) \oplus \mathbf{r}_i$ is the trivial representation.[12]

We next embed $G$ as a subgroup of $G'$, $\pi : G \hookrightarrow G'$, chosen such that there is no global anomaly in $G'$. This embedding induces an injection $\pi_*$ from the space of spin 5-manifolds with $G$-bundles, $\mathcal{M}(G)$, to the space of spin 5-manifolds with $G'$-bundles, $\mathcal{M}(G')$. More explicitly, since $G$ is a subgroup of $G'$, we can view $A$ as a background gauge field for $G'$ (albeit not the most general background for $G'$), which we will denote by $A'$. In other words,

$$(Y_{\text{tot}}, A') = \pi_*(Y_{\text{tot}}, A). \tag{A3}$$

This, in turn, induces a pullback on the space of homomorphisms $\mathcal{M}(G) \rightarrow \mathbb{R}$:

$$\pi^* : \text{Hom}(\mathcal{M}(G'), \mathbb{R}) \rightarrow \text{Hom}(\mathcal{M}(G), \mathbb{R}) :$$
$$f \mapsto \pi^* f = f \circ \pi_*. \tag{A4}$$

Similarly, $\pi$ induces a pullback on the representation ring:

$$\pi^* : RU(G') \rightarrow RU(G) :$$
$$\mathbf{r}' \mapsto \pi^* \mathbf{r}'. \tag{A5}$$

---

[12] Physically, one can think of $\eta_{-\mathbf{r}_i}$ as evaluated for the fermion in the irrep $\mathbf{r}_i$ of $G$ in the opposite chirality.

where $\pi^*\mathbf{r}'$ is $\mathbf{r}'$ decomposed into direct sums of irreps of the subgroup $G$. Two $\eta$-invariants, $\eta : \mathcal{M}(G) \to \mathbb{R}$ and $\eta' : \mathcal{M}(G') \to \mathbb{R}$, are related to each other by the naturality property of the $\eta$-invariant [95, 96], that is, the diagram

$$
\begin{array}{ccc}
RU(G) & \xrightarrow{\ \ \eta\ \ } & \mathrm{Hom}\,(\mathcal{M}(G), \mathbb{R}) \\
\pi^* \Big\uparrow & & \Big\uparrow \pi^* \\
RU(G') & \xrightarrow{\ \ \eta'\ \ } & \mathrm{Hom}\,(\mathcal{M}(G'), \mathbb{R})
\end{array}
\tag{A6}
$$

is commutative. Equivalently, it means

$$
\eta_{\pi^*\mathbf{r}'} = \pi^*\eta'_{\mathbf{r}'} = \eta'_{\mathbf{r}'} \circ \pi_* \,.
\tag{A7}
$$

In categorical terms, we say that $\eta$ is a natural transformation from the functor $RU(\bullet)$ to the functor $\mathrm{Hom}\,(\mathcal{M}(\bullet), \mathbb{R})$, where both are functors from the category of groups to the category of abelian groups.

We can now use the naturality of the $\eta$-invariant to rewrite the global anomaly in $G$ as a perturbative anomaly in $G'$. Using Eq. (A7), the partition function for the invertible theory $\mathcal{I}$ on $(Y_\mathrm{tot}, A)$ corresponding to free fermions in the representation $\mathbf{r}$ of $G$ can be evaluated as

$$
Z_\mathcal{I}[Y_\mathrm{tot}; A] = \exp\left(-2\pi i \eta_\mathbf{r}(Y_\mathrm{tot}, A)\right) = \exp\left(-2\pi i \eta'_{\mathbf{r}'} \circ \pi_*(Y_\mathrm{tot}, A)\right)
\tag{A8}
$$

$$
= \exp\left(-2\pi i \eta'_{\mathbf{r}'}(Y_\mathrm{tot}, A')\right) \,,
\tag{A9}
$$

where we pick $\mathbf{r}'$ such that $\pi^*\mathbf{r}' = \mathbf{r}$. Recall that $G'$ is chosen such that there are no global anomalies, i.e. $\Omega_5^\mathrm{Spin}(BG') = 0$.[13] This means $(Y_\mathrm{tot}, A')$ can be extended as a boundary of a 6-manifold $W$, with all the structures appropriately extended. In particular, $W$ is equipped with a $G'$-bundle with background gauge field $\tilde{A}'$ such that $\tilde{A}'|_\partial = A'$. This extension is only possible because we allow a more general $G'$-bundle instead of restricting to $G$-bundles over the bulk $W$. Since $(Y_\mathrm{tot}, A') = \partial(W, \tilde{A}')$, we can apply the APS index theorem to express the $\eta$-invariant on $(Y_\mathrm{tot}, A')$ in terms of the anomaly polynomial,

$$
Z_\mathcal{I}[Y_{tot}; A] = \exp\left(-2\pi i \eta'_{\mathbf{r}'}(Y_\mathrm{tot}, A')\right) = \exp\left(-2\pi i \int_W \hat{A}(R)\mathrm{tr}_{\mathbf{r}'}\left[\exp\left(\frac{\tilde{F}'}{2\pi}\right)\right]\right) ,
\tag{A10}
$$

where $\tilde{F}'$ is the field strength of $\tilde{A}'$ on $W$. This expression provides the sought-after differential form version of global anomalies that we can then apply to analyses of anomalies in hydrodynamics.

It is worth remarking that, in order to have the differential form expression (A10) for the exponentiated $\eta$-invariant, we do not strictly require that $\Omega_5^\mathrm{Spin}(BG') = 0$. This is rather a convenient shortcut which, when available, implies that any closed $(Y_{tot}, A')$ *must* be null-bordant. But even if $\Omega_5^\mathrm{Spin}(BG') \neq 0$, if we can nonetheless construct an explicit extension $(W, \tilde{A}')$ such that $(Y_\mathrm{tot}, A') = \partial(W, \tilde{A}')$, we can of course use the APS index theorem to arrive at (A10).

---

[13] Similar ideas, exploiting the naturality of the $\eta$-invariant, were used in [94] to relate global anomalies associated with non-abelian finite groups to other global anomalies, for instance in abelian finite groups.

**Application to the Witten anomaly at finite temperature**

We now apply this formalism to offer an alternative calculation for the $\eta$-invariant pertaining to the Witten anomaly, adapted to the hydrodynamic setting as in §IV A. This is a more pedestrian way to evaluate $\eta(S_\beta^1 \times S^4)$ than the factorization property (4.10) that we invoked in the main text, and is in essence reproduced from the calculation in [93].

Consider the $SU(2)$ gauge field configuration on $S_\beta^1 \times S^4$ with holonomy $\mu/T$ around the thermal cycle $S_\beta^1$, which just corresponds to the system being at finite chemical potential, and furthermore twisted by $(-1)^F$, which coincides with the central element in $\mathcal{Z}(SU(2)) \cong \mathbb{Z}_2$. Recall the gauge field on the $S^4$ factor is in an instanton configuration, obtained by gluing two hemispheres via the clutching construction $A_{\text{clutch}}$. In terms of patches, we can express $A_{\text{clutch}} = \mathcal{A}$ on the upper hemisphere and $A_{\text{clutch}} = \mathcal{A}^g$ on the lower hemisphere respectively.

This configuration of $G = SU(2)$ can be embedded inside $G' = U(2) \cong (SU(2) \times U(1))/\mathbb{Z}_2$, for which $\Omega_5^{\text{Spin}}(BU(2)) = 0$ [54, 97, 98], as

$$A' = -\tilde{\eta}\pi T u \mathbb{1} - \mu u + A_{\text{clutch}} \,. \tag{A11}$$

The first term is a component of the gauge field in the direction $U(1) \subset U(2)$ by which we centrally extend the original $SU(2)$ to get $U(2)$. We see that the charge $\tilde{\eta} \in \mathbb{Z}_{\text{odd}}$ corresponds to the centre element that results in the fundamental fermion gaining phase $e^{i\pi}$ as it goes around the thermal cycle, i.e.

$$\exp\left(i\int_{S_\beta^1} A'\right)\bigg|_{\mu=0} = -\mathbb{1} \,. \tag{A12}$$

This $U(1) \subset U(2)$ charge $\tilde{\eta}$ is moreover related to the isospin $j$ of the microscopic fermion representation, by the 'isospin-charge' relation

$$\tilde{\eta} = 2j \mod 2 \,. \tag{A13}$$

Since $\Omega_5^{\text{Spin}}(BU(2)) = 0$, one can evaluate the $\eta$-invariant via the 6d anomaly polynomial on $W$ with $W = D^2 \times S^4$ via the APS index theorem. We choose the extension of $A'$ to $A'(r)$ on $W$ to be

$$A'(r) = -(\tilde{\eta}\pi T u \mathbb{1} - \mu u)r + A_{\text{clutch}} \tag{A14}$$

such that the gauge field on $r = 1$ corresponds to that on $\partial W = S_\beta^1 \times S^4$. Then

$$\begin{aligned}
\eta(S_\beta^1 \times S^4) &= \frac{1}{3!}\frac{1}{(2\pi)^3}\int_{D^2 \times S^4} \text{tr}\left[F'(r)^3\right] \,, \\
&= -\frac{\tilde{\eta}}{16\pi^2}\int_{r=0}^{r=1} dr \int_{S_\beta^1} uT \int_{S^4} \text{tr}(F_{\text{clutch}} \wedge F_{\text{clutch}}) \,, \\
&= -\frac{1}{2}\tilde{\eta}T(j)\int_{S^4} p_1(F_{\text{clutch}}) \,.
\end{aligned} \tag{A15}$$

Here, $T(j) = \frac{2}{3}j(j+1)(2j+1)$ is the Dynkin index of the representation $j$ as in the main text. To relate this to the result in the main text, the value of the $U(1)$ charge $\tilde{\eta}$ plays the role of $2\eta(S^1_\beta)$, whose values agree modulo 2. It is non-trivial only when the spin structure flips from AP to P upon the centre insertion for $j \in \mathbb{Z}_{\text{odd}}$. Had we turned off the defect operator, $U_\pi$ or $(-1)^F$ in (4.13), there will be no anomalous phase.

Similar to how we embedded $SU(2)$ inside $U(2)$, one can embed the group $USp(2N)$ inside

$$G' = \frac{USp(2N) \times U(1)}{\mathbb{Z}_2}. \tag{A16}$$

The $\mathbb{Z}_2$ quotient relates the $U(1)$ charge and the analogue of 'isospin' of the representation of $USp(2N)$. One can then apply the anomaly interplay procedure to express the exponentiated $\eta$-invariant of the mapping sphere, even though we have not actually evaluated the bordism groups $\Omega_5^{\text{Spin}}(BG')$. In this more general case, one can still use the $U(1)$ direction in the extended gauge group to revert back to AP spin structure on $S^1_\beta$ which can be explicitly filled in to a 2-disc in spin bordism, again with a $U(1)$ monopole field at the centre of that disc whose charge is fixed modulo 2 by the Dynkin labels of the $USp(2N)$ representation (and in particular whether the central $\mathbb{Z}_2 \subset \mathcal{Z}(USp(2N))$ element is non-trivial in that representation). The calculation then proceeds as for the $SU(2)$ case.

---

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
