# Peer review of "Fractional Hydrodynamic Transport from the Witten Anomaly"

_SciPost Physics_

## Round 1 · Referee Report · Anonymous (Referee 1) · 2025-5-22

Strengths

  1. A novel observation of a potentially interesting effect

Weaknesses

  1. The paper is is written in a very formal way
  2. It is not clear it the effect is physical or it is just a mathematical curiosity

Report

The authors ask a question: can a global anomaly survive in hydrodynamics?

At first glance, it might seem that a global anomaly, such as the SU(2) Witten anomaly, should not influence hydrodynamic behavior. Hydrodynamic effective field theories are constructed from local, continuous degrees of freedom, while global anomalies are discrete (specifically $\mathbb{Z}_2$-valued) and non-perturbative in nature. Since global anomalies do not lead to local current non-conservation, one might assume they are irrelevant in hydrodynamic regimes.

However, the authors demonstrate that this assumption is incorrect. By employing anomaly inflow mechanisms and bordism theory, the authors show that the anomaly does in fact contribute to the three-dimensional effective theory obtained by thermal dimensional reduction. Specifically, it manifests as a fractional-level non-abelian Chern--Simons term, which influences physical observables.

To detect the presence of the global anomaly in thermal equilibrium, the authors consider a spacetime geometry of the form
\[
X_4 = S^1_\beta \times S^3,
\]
where $S^1_\beta$ is the thermal circle and $S^3$ is a spatial three-sphere.

They construct a five-dimensional manifold (bulk)
\[
Y_5 = S^1_\beta \times D^4,
\]
such that its boundary is the physical spacetime $X_4$.

To expose the anomaly, the authors glue two copies of this bulk together along $X_4$ using a non-trivial SU(2) gauge transformation $g: S^3 \rightarrow SU(2)$. This procedure, called \emph{clutching}, results in a five-dimensional mapping sphere,
\[
S^1_\beta \times S^4,
\]
equipped with a non-trivial SU(2) gauge bundle determined by the topological class of $g$.

The anomaly is detected via the evaluation of the $\eta$-invariant on this configuration. A non-trivial value of this invariant signals the presence of the global anomaly.

In the dimensionally reduced three-dimensional theory, the presence of the global anomaly implies that the effective action must include a fractional Chern--Simons term:
\[
\log Z \supset \frac{q(j)}{8\pi} \int \mathrm{CS}_3[A],
\]
where $q(j) = T(j) \mod 2$, and $T(j)$ is the Dynkin index of the SU(2) representation. This term leads to fractionally quantized transport coefficients in response to background magnetic fields and vorticity, demonstrating that the global anomaly contributes to hydrodynamic transport, albeit in a subtle and topologically protected manner.

I have two concerns regarding the construction in the mansucript:
\begin{itemize}
\item In the analysis, compatibility with thermal equilibrium is taken to require that both the background gauge field on \( X_4 = S^1_\beta \times M_3 \) and any gauge transformation \( g \) be independent of the Euclidean time (thermal circle) direction. This condition is necessary to ensure a well-defined dimensional reduction to a three-dimensional effective theory.

However, it may be useful to clarify whether additional assumptions are implicitly made in this restriction. For instance, if the system carries angular momentum or is placed in a rotating frame, then the thermal circle may participate non-trivially in spacetime symmetries. In such contexts, it is conceivable that gauge transformations could acquire dependence on the thermal cycle coordinate.

This raises the question of whether it is always valid to enforce strict independence from the thermal direction, or whether relaxing this assumption might reveal additional topological or anomaly-related effects. Moreover, the independence seems to be important in the construction.

\item A further conceptual point concerns the interpretation of the so-called \emph{twisted thermal state}, which arises when fermions are assigned periodic boundary conditions around the Euclidean thermal circle. In standard thermal field theory, physical thermal equilibrium requires that fermions be anti-periodic on \( S^1_\beta \) to ensure the correct Fermi–Dirac statistics and thermal ensemble:
\[
Z_{\text{thermal}} = \mathrm{Tr}(e^{-\beta H}).
\]
By contrast, assigning periodic boundary conditions yields
\[
Z_{\text{twisted}} = \mathrm{Tr}((-1)^F e^{-\beta H}),
\]
which computes the Witten index and counts net zero-energy states. Such a configuration is not realized by any physical heat bath, nor does it describe a thermally equilibrated system in the usual sense. Therefore, the twisted thermal state is not physical in a thermodynamic sense.

This raises a point for clarification: If one goes to the lab can the authors' claim that non-perturbative anomalies can be measured be verified or not? If not, which seems to be the case, I would not claim the connection to transport.

\end{itemize}

Attachment

Recommendation

Ask for major revision

---

## Round 1 · Referee Report · Anonymous (Referee 2) · 2025-10-1

Strengths

  1. A potentially new application of a rather abstract formulation of anomalies.

Weaknesses

  1. Concrete observable effects are not clear.

Report

I have the same question as Referee 1. The boundary condition in the thermal direction is periodic rather than antiperiodic. It is not clear how to realize that boundary condition in actual experiments. (The usual antiperiodic boundary condition is realized by finite temperature).

Another comment is as follows. Under the periodic boundary condition, there is no mass gap in 3 dimensions if we consider e.g. free fermions (with SU(2) symmetry). There are some massless fermions in 3d, and they have parity anomaly. The parity anomaly in 3d can match the Witten SU(2) anomaly in 4d. In particular, notice that the (Euclidean version of the) CPT transformation in 4d becomes the parity transformation in 3d, and the parity anomaly is a mixed anomaly between gauge symmetries and parity transformation.

Requested changes

  1. More discussions on the boundary condition (whether it can be realized experimentally) should be given.

  2. The existence of massless fermions under the periodic boundary condition and their parity anomaly should be mentioned.

Recommendation

Ask for major revision

---

## Editorial Decision

awaiting_resubmission